# Design of a Measuring Device and Winch Structure for Detecting the Distance and Direction of Two Seabed Pipelines

**Zhuo Wang [1,\*], Di Lan [2], Tao Wang [3] and Bo Zhang [1]**

[1] College of Mechanical and Electrical Engineering, Harbin Engineering University, Harbin 150001, China; zhangbo_heu@hrbeu.edu.cn

[2] Technische Universität Dresden, Helmholtzstr.10, 01069 Dresden, Germany; landirobin@outlook.com

[3] School of Mechanical Engineering, Hebei University of Technology, Tianjin 300130, China; 18846166436@hrbeu.edu.cn

[\*] Correspondence: wangzhuo_heu@hrbeu.edu.cn

**Abstract:** To measure the distance and direction between the flanges of two seabed pipelines, a measuring device for pulling a rope in seawater was designed. Addressing the sealing problem of the key equipment the rotating shaft of the rope winch, we used the magnetic coupling principle to transfer the driving moment, and converted the dynamic seal into a static seal structure to reliably seal the motor. Through an experiment measuring two pipelines with the underwater rope pulling device, we verified that the measuring accuracy of the device meets the design requirements, and confirmed the feasibility of applying magnetic coupling technology in winches.

**Keywords:** undersea project; measuring two pipelines; rotating shaft; magnetic coupling; assisted rope winch

## 1. Introduction

As the depth of marine development increases, connections of two pipelines on the seafloor are required for the development of deep-sea oil and gas resources. To complete the tie-back operation between both connecting pipelines, the relative distance and the pose of the center of two pipe flanges must be measured. Using an auxiliary tension rope, a deep-water pipeline orientation measurement device measures the relative spatial direction and submarine azimuth angle between two pipeline flanges [1]. As an important part of the complete machine, the main function of the winch is to store, retract, and tension the measuring rope [2]. The motor is an important driving element of the underwater winch. In ultra-deep-water environments, which is more than 1500 meters, dynamically sealing the rotating shaft of the winch bottleneck is the problem [3]. This paper presents a new type of underwater winch structure based on magnetic coupling technology.

The four common dynamic sealing methods are (1) O-ring seals; (2) slip ring combined seals, which combines O-ring seal with PTFE (Polytetrafluoroethylene) or some other slip rings [4]; (3) mechanical seals. The CAS (Chinese Academy of Sciences) Shenyang Institute of Automation proposed a kind of mechanical seal structure, through which the appliance achieved good leak-proof effects of 10 MPa with a rotating shaft speed of 1150 rpm [5]. The 705 Institute of CSIC China (Shipbuilding Industry Corporation) proposed a mechanical seal construction applied to the tail shaft of a torpedo. As such, the torpedo could bear a pressure of 4.5 MPa with a maximum speed of 3200 rpm [6]. This method has been successfully applied in the China CR 01 6000 m underwater robot. (4) The last sealing method is magnetic fluid seals. As cutting-edge research, their application is mature in vacuum low-pressure seals [7]. More recently, they have gradually been applied in liquid seals, including marine screw

propeller seals, such as magnetic fluid seals of marine tail shafts based on $Fe_3O_4$ [8], as well as medical instrument seals for blood pumps, such as the micro-magnetic fluid sealing system based in a liquid environment [9].

Generally, O-ring seals can withstand rotational dynamic pressure less than 1.5 MPa with a line speed of the rotating shaft of 3.5 m/s. However, this method has some deficiencies: its large starting friction resistance easily produces the crawling phenomenon. When the rotating shaft rotates with high speed, the friction heat accelerates the aging process and the friction results in grooves on the shaft, which can cause the leakage [10]. Slip ring combined seals support pressures under 60 MPa and a line speed of the rotating shaft within 6 m/s [11]. However, this type of dynamic seal method experiences abrasion. As such, the wearing slip rings need to be replaced regularly, and the installation process is relatively complex [12]. Mechanical seals can be applied under a maximum pressure of 45 MPa and a line speed of 100 m/s. These seals have a complex structure and the cost is comparatively high [13]. The pressure-bearing capacity of magnetic fluid seals is relatively low. The single-stage pressure-bearing capacity is below 120–180 KPa and the overall capacity rarely exceeds 11.5 MPa. When magnetic fluid is used to seal the liquid, its durability decreases to some extent due to diffusion, especially in deep-water environments [14].

Seal technology is divided into dynamic and static seals. The static seal structure is relatively simple and the pressure-bearing can reach 150 MPa without wear problems. Given the above-mentioned findings and based on converting the dynamic seal with an assisted rope winch into a static seal [15], this paper presents a sealing method using a magnetic coupling coupler. The magnetic driven rotor is driven by the active magnetic rotor, which is actuated by the motor. The fabric achieves contactless transmission and is leak-proof. The sealing method provides overload protection for the motor [16].

## 2. Design of the Structure of the Measuring Device and Winch

### 2.1. Structure of Measuring Device

The measuring device is shown in Figure 1. It mainly consists of a measurement section, butting base, and a winch.

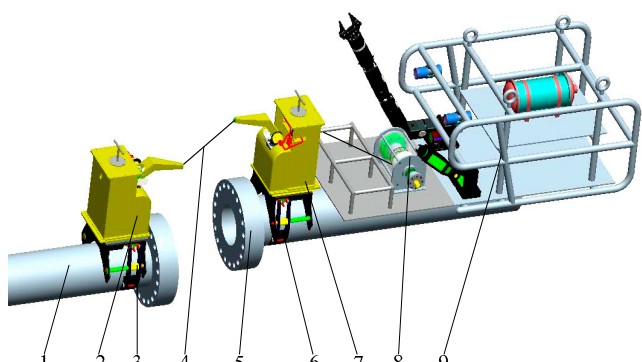

**Figure 1.** The deep-water pipeline pose measurement device. 1, pipeline 1; 2, measurement section I; 3, butting base I; 4, detection rope; 5, pipeline 2; 6, butting base II; 7, measurement section II; 8, winch; 9, ROV (Remote Operated Vehicle).

The measurement section is a platform composed of detection sensors, used to detect the horizontal swing angle $\gamma$ and vertical pitch angle $\theta$ through the extension arm. The orthogonal internal inclination sensor determines the horizontal and vertical tilt angle $\alpha$, $\beta$, which are relative to the docking plane. The internal detection mechanism of the rope length measures the tension length $L$ of the detection rope. Based on the detection data, the measurement model is shown in Figure 2.

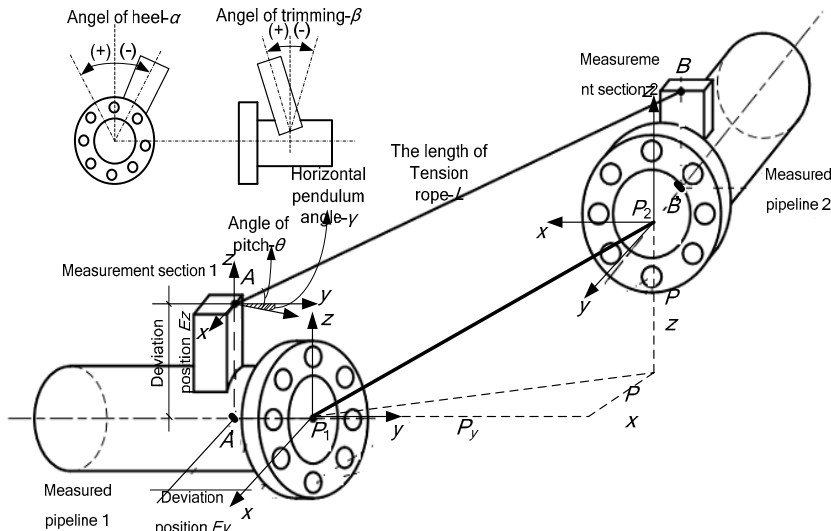

**Figure 2.** The parameters of relative pipeline poses.

As the winch performs collection and tension and other rope detection actions during underwater measurement, the performance of the apparatus is vastly influenced by the winch. Therefore, studying the tightness of the detection rope and the sealing method under deep water is necessary.

The required data are shown in Figure 2, where $\alpha_r$ is the angle between measurement section I and the vertical plane, $\beta_r$ is the angle between measurement section I and the horizontal plane, $\gamma_r$ is the horizontal swing angle between measurement section I and the extension arm, $\theta_r$ is the pitch angle between measurement section I and the extension arm, $\alpha_b$ is the angle between measurement section II and the vertical plane, $\beta_b$ is the angle between measurement section II and the horizontal plane, $\gamma_b$ is the horizontal swing angle between measurement section II and the extension arm, $\theta_b$ is the pitch angle between measurement section II and the extension arm, and $\varphi_L$ is the rotation angle of the detection mechanism.

### 2.2. Analysis of Tension State of the Detection Rope

Due to the distance between the two pipes and the external forces, guaranteeing a straight line rope measurement in a strict sense is difficult. The rope is a drooping curve due to its own weight and the external load. The pulling force is the largest when the detection rope is tensioned. Therefore, the tension force is the basis of the motor's output torque. According to the design requirements, the maximum measuring distance is 30 m. One end of the test rope was fixed on the bracket; the other end was connected to the spring scale and placed in another bracket. Both brackets were the same height. The distance between both brackets was adjusted to 5, 10, 20 and 30 m, and we determine the required tension for different spacings using a spring dynamometer, as shown in Table 1.

**Table 1.** The required tension for different sags.

| Tension $F_T$ (N) | | Spacing $l$ (m) | | | |
|---|---|---|---|---|---|
| | | 5 | 10 | 20 | 30 |
| **Sag $f$ (m)** | 0 | 30 | 80 | 230 | 380 |
| | 0.01 | 25.7 | 67.9 | 214.9 | 361.6 |
| | 0.05 | 13.9 | 38.3 | 176.9 | 309.6 |
| | 0.1 | 7.5 | 30.6 | 124.3 | 254.4 |
| | 0.2 | 3.9 | 15.4 | 61.3 | 173.2 |
| | 0.3 | 2.6 | 10.3 | 40.9 | 121.8 |

The winch motor's drive torque was calculated according to the diameter $D$ of the drum and the transmission efficiency $\eta$ of the winch. In this paper, $D$ was 0.2 m and $\eta$ was 0.9. According to design requirements, the relative error of the measurement of the length of the rope is under 0.01%. We determined that the drive torque was 24 Nm. Considering the motor chamber was small, to improve the durability [17–19], we selected a brushless DC motor BG62 × 60 SNR88562 01571 3880 RPM and reducer products (Dunkermotoren, Bonndorf, Germany). The relevant parameters are shown in Table 2.

**Table 2.** Main dimensions and performance parameters of the motor.

| Parameter | Value |
| --- | --- |
| Weight | 3 kg |
| Torque | 24 Nm |
| Voltage | 24 V |
| Electricity | 6.44 A |
| Rotation rate | 50 rpm |
| Shaft diameter | 15 mm |

### 2.3. Total Winch Design Scheme

The total winch design is shown in Figure 3. The overall framework of the winch included locating rods (part 3), a left support plate (part 1), and a right support plate (part 8), which were connected to part 3 with bolts. Via bolt fastening, the drum consisted of a left drum cover (part 2), a barrel (part 4), and a right drum cover (part 7). The motor (part 16) was fixed in the hermetically sealed enclosure (part 15) with a left-hand flange that was magnetically coupled to the element (part 11) by bolting. The O-ring (part 14) was installed between the two flanges to form a static seal structure. The outer magnetic rotor (part 10) was connected to the left drum cover (part 2) to form a rotary assembly using a key joint (part 9). The hermetically sealed enclosure (part 15) was connected to the right support plate (part 8) by a key (part 17) to create a motor fixed assembly. The right end of the motor (part 16) was formed by the right cover (part 18) of the motor through the O-ring to form the static seal assembly. The dynamic seal was transformed into the static seal structure.

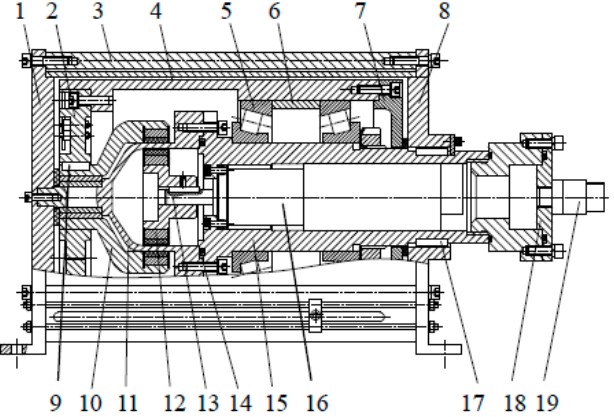

**Figure 3.** Schematic diagram of magnetic coupling winch. 1, left support plate; 2, left drum cover; 3, locating rods; 4, barrel; 5, roller bears; 6, spacer sleeve; 7, right drum cover; 8, right support plate; 9, key joint; 10, outer magnetic rotor; 11, left-hand flange; 12, active magnetic rotor; 13, motor shaft; 14, O-shaped seal; 15, sealed enclosure; 16, motor; 17, fixing key; 18, right cover; 19, watertight cable glands.

As showed in Figure 3, the magnetic coupling coupler consisted of an active magnetic rotor (part 12), a driven magnetic rotor (part 10), and an isolation sleeve (part 11) provided in the working gap of the magnetic rotor [20]. The isolation sleeve was used to seal the inner magnetic circuit rotor and motor. To produce enough magnetic force, the distance between the inner and outer magnetic rotors should be close. Thus, the isolation sleeve material should have higher strength and toughness, and the magnetic permeability $\mu \approx 1$. In this study, we selected the austenitic stainless steel-18Cr-12Ni-2.5Mo (316L). To prevent blocking the magnetic field, the calculation formula for the thickness of the isolation sleeve is as follows [21]:

$$\delta_t = \frac{pD_n}{2[\sigma_b]\varphi - p} \tag{1}$$

where $\delta_t$ is the thickness of the pressure shell (mm), $p$ is the design pressure (MPa), $D_n$ is the inner diameter (mm), $[\sigma_b]$ is the allowable stress (MPa) and $\Phi$ is the welding coefficient. In this paper, $\varphi = 1$.

The isolation sleeve is trans-mutative under pressure. In this study, we mainly considered the radial deformation:

$$\Delta S = \frac{pR_1^2}{ES(1 - \frac{\mu}{2})} \tag{2}$$

where $\Delta S$ is the radial deformation (mm), $p$ is the design pressure (MPa), $R_1$ is the inner diameter of the isolation sleeve (mm), $E$ is the modulus of elasticity (MPa), $S$ is the thickness of the isolation sleeve (mm), and $\mu$ is the Poisson ratio.

Considering the margin of strength, the external pressure $p$ was taken as 18 MPa and $D_n$ as 90 mm. According to Equations (1) and (2), the calculation results were $\delta_t = 4.7$ mm and $\Delta S = 0.1$ mm. So, we used an isolation sleeve thickness of 5 mm.

The torque between the inside and outside magnetic rotor can be calculated using

$$T = \left(\frac{1}{5000}\right)^2 KMHmSt_h R_c \sin(\frac{m}{2}\phi) \tag{3}$$

where

$$M = \frac{B_m + H_m}{4\pi}$$

$$H = N_1 \times 4\pi m(1 - \frac{t_g}{\sqrt{t_g^2 + t_0^2}})\eta$$

$$t_h = \frac{1}{2}(t_{im} + t_{om})$$

$$R_c = \frac{1}{2}(R_2 + R_3)$$

where $T$ is torque, kgf·cm; $K$ is the magnetic circuit coefficient, $K = 4$–6.4; $M$ is the intensity of magnetization (Gs); $B_m$ and $H_m$ are the magnetic induction and the magnetic field strength of the working points, respectively; $H$ is the magnetic field strength generated by the outer magnetic circuit at the inner magnetic circuit (Oe); $N_1$ is the empirical coefficient of the shape of the magnetic pole, where the fan-shaped pole $N_1 = 1.05$ and rectangular and square pole $N_1 = 1.24$; $t_g$ is the width of the working gap (cm); $t_0$ is the length of the magnetic pole, cm; $\eta$ is the magnet thickness coefficient, correlated with $t_h/t_0$; $m$ is the number of the pole; $S$ is the magnet pole acreage (cm$^2$); $t_h$ is the magnet thickness (cm); $R_c$ is the average rotational radius to the center of rotation of the magnetic force applied to the inner magnetic pole (cm); $\varphi$ is the displacement angle during operation.

Equations (2) and (3) reflect that the transmission torque $T$ is proportional to the magnetic energy product $BH$, changing with the magnetic rotation angle. As the magnetic energy product of the permanent magnet is mainly determined by the material, we selected Nd-Fe-B, which has high remanence, high coercive force and high magnetic energy.

In the design, the maximum torque is 16 Nm. Based on the maximum torque, the sizes of the internal and external rotors are shown in Figure 4. The flange of the motor chamber that serves to seal the motor and the control system is connected to the isolation sleeve of the magnetic coupler. Its structure type is shown in Figure 5.

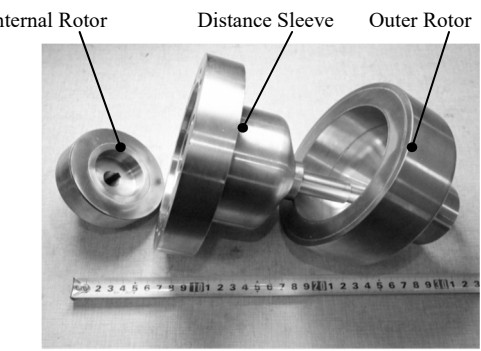

**Figure 4.** Photo of the magnetic coupling coupler.

The tail uses spiral sealing and the electronic chamber uses a pressed seal for the installation of the motor control circuit. Aluminum alloy was selected as the motor chamber material to reduce the global mass. The thickness of the motor chamber was determined to be 10.5 mm according to the Equation (1). However, considering its poor corrosion resistance, in this study, the thickness was set to 15 mm.

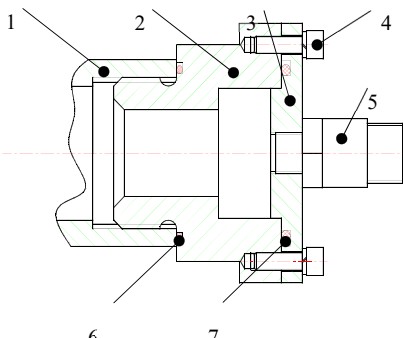

**Figure 5.** The structure and sealing style of the motor chamber. 1, motor chamber; 2, electronic storehouse; 3, end cover; 4, set screw; 5, watertight joint; 6, first seal; 7, second seal.

## 3. Motor Control Simulation and Rope Pulling Force Experiments

The machine winding was connected in star type, and its basic structure is shown in Figure 6.

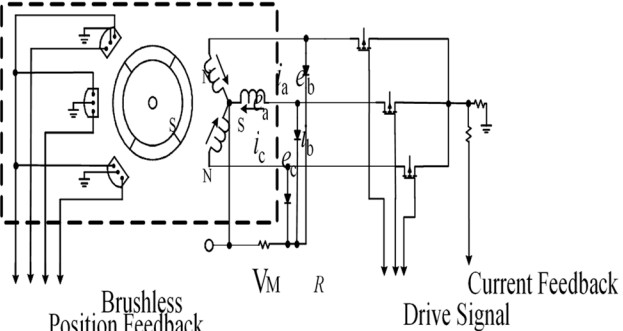

**Figure 6.** Star type connection of brushless motor.

We established the mathematical model of the brushless motor [22–24]:

$$
\begin{bmatrix} v_a \\ v_b \\ v_c \end{bmatrix} = \begin{bmatrix} R & 0 & 0 \\ 0 & R & 0 \\ 0 & 0 & R \end{bmatrix} \begin{bmatrix} i_a \\ i_b \\ i_c \end{bmatrix} + p \begin{bmatrix} L_a & L_{ba} & L_{ca} \\ L_{ba} & L_b & L_{cb} \\ L_{ca} & L_{cb} & L_c \end{bmatrix} \begin{bmatrix} i_a \\ i_b \\ i_c \end{bmatrix} + \begin{bmatrix} e_a \\ e_b \\ e_c \end{bmatrix}
\tag{4}
$$

where $v_a$ $v_b$ $v_c$ are stator winding phase voltage in V, $i_a$ $i_b$ $i_c$ are stator winding phase current in A, $e_a$ $e_b$ $e_c$ are the stator winding opposite electromotive force in V, $R$ is the resistance of each phase winding in Ω, $L_a$ $L_b$ $L_c$ are the inductance of each phase winding in H, and $L_{ab}$ $L_{bc}$ $L_{ca}$ are the mutual inductance of each two-phase windings in H. $L_a = L_b = L_c = L$, $L_{ab} = L_{bc} = L_{ca} = M$, and $i_a + i_b + i_c = 0$.

Thus,

$$
\begin{bmatrix} v_a \\ v_b \\ v_c \end{bmatrix} = \begin{bmatrix} R & 0 & 0 \\ 0 & R & 0 \\ 0 & 0 & R \end{bmatrix} \begin{bmatrix} i_a \\ i_b \\ i_c \end{bmatrix} + p \begin{bmatrix} L & M & M \\ M & L & M \\ M & M & L \end{bmatrix} \begin{bmatrix} i_a \\ i_b \\ i_c \end{bmatrix} + \begin{bmatrix} e_a \\ e_b \\ e_c \end{bmatrix}
\tag{5}
$$

$$
\begin{bmatrix} v_a \\ v_b \\ v_c \end{bmatrix} = \begin{bmatrix} R & 0 & 0 \\ 0 & R & 0 \\ 0 & 0 & R \end{bmatrix} \begin{bmatrix} i_a \\ i_b \\ i_c \end{bmatrix} + \begin{bmatrix} L-M & 0 & 0 \\ 0 & L-M & 0 \\ 0 & 0 & L-M \end{bmatrix} p \begin{bmatrix} i_a \\ i_b \\ i_c \end{bmatrix} + \begin{bmatrix} e_a \\ e_b \\ e_c \end{bmatrix},
\tag{6}
$$

$$
p \begin{bmatrix} i_a \\ i_b \\ i_c \end{bmatrix} = \begin{bmatrix} 1/(L-M) & 0 & 0 \\ 0 & 1/(L-M) & 0 \\ 0 & 0 & 1/(L-M) \end{bmatrix} \left[ \begin{bmatrix} v_a \\ v_b \\ v_c \end{bmatrix} - \begin{bmatrix} R & 0 & 0 \\ 0 & R & 0 \\ 0 & 0 & R \end{bmatrix} \begin{bmatrix} i_a \\ i_b \\ i_c \end{bmatrix} - \begin{bmatrix} e_a \\ e_b \\ e_c \end{bmatrix} \right].
\tag{7}
$$

Based on the motor torque equation,

$$
T_e = (e_a i_a + e_b i_b + e_c i_c)/\omega_r
\tag{8}
$$

we obtained

$$
p\omega_r = (T_e - T_L - B\omega_r)/J
\tag{9}
$$

Based on the above formula derivation, we built the brushless motor drive model in the MATLAB/Simulink environment as shown in Figure 7 [25,26].

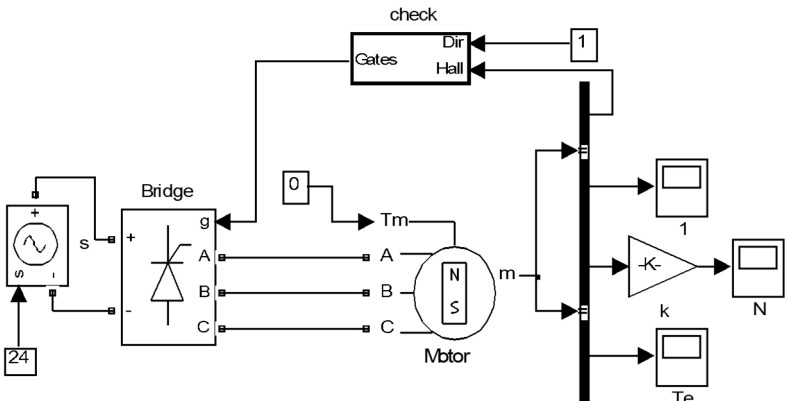

**Figure 7.** MATLAB/Simulink model of brushless motor.

We established the motor servo control system based on Figure 7, as shown in Figure 8. The servo system was a closed-loop system consisting of a speed loop and a position loop. The speed loop adopted a PI (proportional-integral) regulator, and the position loop, a PD (proportional-derivative) regulator.

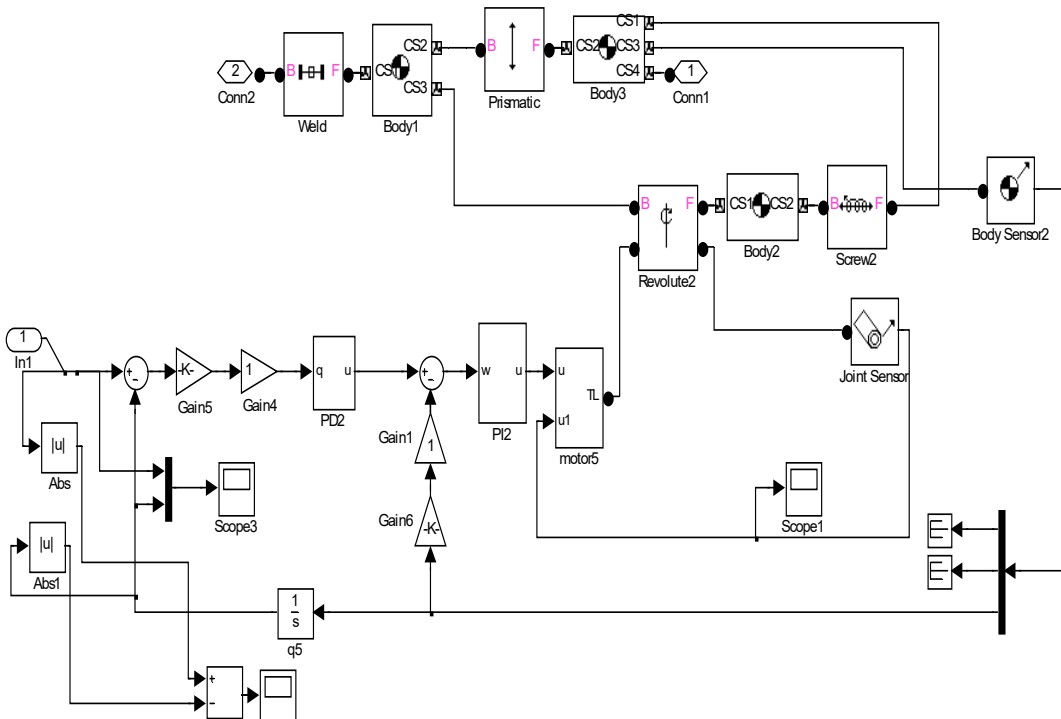

**Figure 8.** The brushless motor servo control system.

On this basis, the closed-loop feedback tracking simulation curve of the motor is shown in Figure 9.

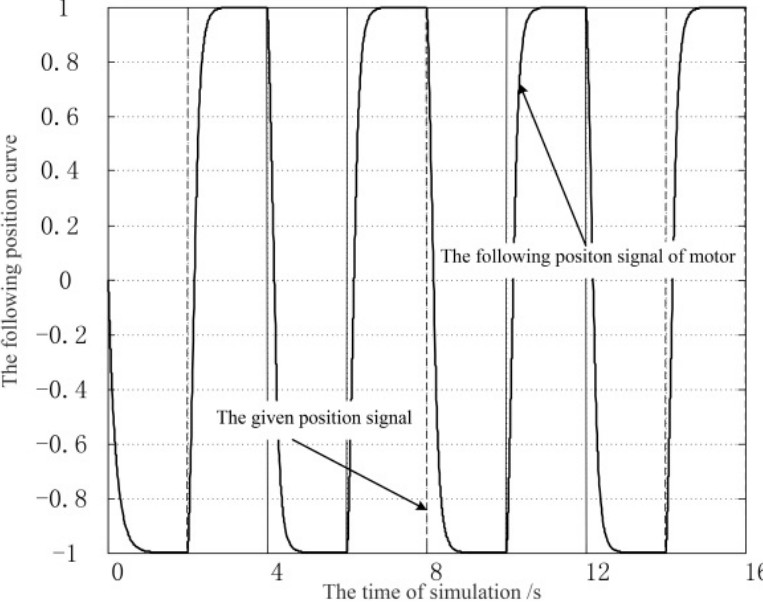

**Figure 9.** The simulation curve of motor.

The overshoot phenomenon causes the cable to loosen from the winding winch, resulting in rope clamping. Therefore, we ensured that the location did not overshoot in the parameter mediation process. The motor has a faster response speed.

Using the simulation results, a motor drive circuit board was designed, as shown in Figure 10. The MC33035 type of Brushless DC Motor Controller for the integrated drive chip was selected as the main driver chip. The Atmgal 8 type of chip was the main control chip, which formed the

communication system with a MAX485 type of chip. The diameter of the circuit board was 60 mm, using a double circuit layout and three MOS tubes drive brushless motor moving.

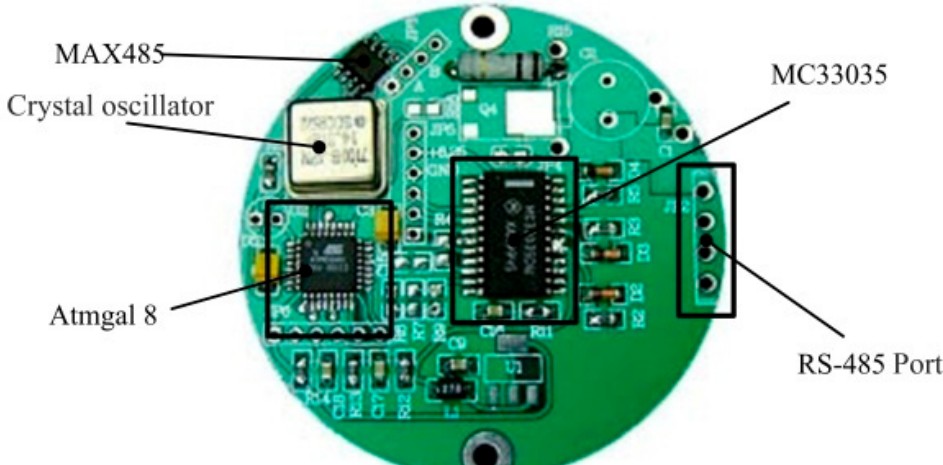

**Figure 10.** The motor drive circuit board.

To test the motor, we first tested the no-load speed control characteristics of the motor. The performance of the motor was stable during the test. The motor was then mounted to the winch for joint commissioning, as shown in Figure 11. The output torque of the commissioning motor was increased from low to high, and the tensile force of the winch was set to 40, 50, 60, 70, 80, 90, 100, 120, and 140 N. The measuring distance was 30 m and the diameter of the pulling rope was 3 mm. we recorded the deflection and deformation of the pulling rope. Table 3 lists the characteristics of the measuring rope under different stretching forces.

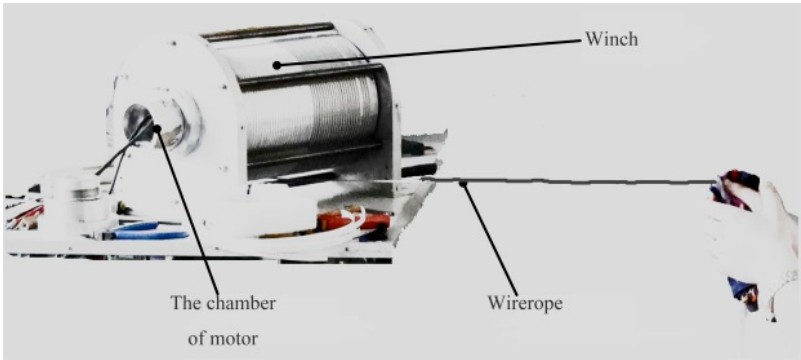

**Figure 11.** The commissioning of servo winch.

**Table 3.** Deflection measurements under different tensile strengths.

| Tensile Force (N) | 40 | 50 | 60 | 70N | 80 | 90 | 100 | 120 | 140 |
|---|---|---|---|---|---|---|---|---|---|
| **Deflection *f* (mm)** | 858.2 | 760.1 | 620.5 | 496.6 | 471.7 | 453.1 | 441.9 | 438.4 | 432.1 |

According to the motor torque value of 24 Nm in Table 2, the maximum tensile force of the inspection rope was obtained from Equation (10):

$$T_N = \frac{F_T D}{2\eta} \tag{10}$$

where $T_N$ is the motor driving torque (nm); $D$ is the diameter of winch drum (m), $D = 0.2$ m; and $\eta$ is the transmission efficiency of the winch, which is mainly the transmission efficiency of the magnetic coupling device, $\eta = 0.9$.

By substituting these data into Equation (10), the maximum tensile force value was calculated as 216 N. The safety margin is 2–2.5, the deflection must be less than 500 mm, and the pulling force was 90 N according to Table 3.

## 4. High-Pressure Experiment

The purpose of this test was to check the high-pressure resistance and the stability of the system.

The experimental device was an 8000 m deep-sea pressure test chamber with an experimental pressure of 80 MPa and an inner diameter of 0.6 m, which can automatically control the experimental process (Figure 12).

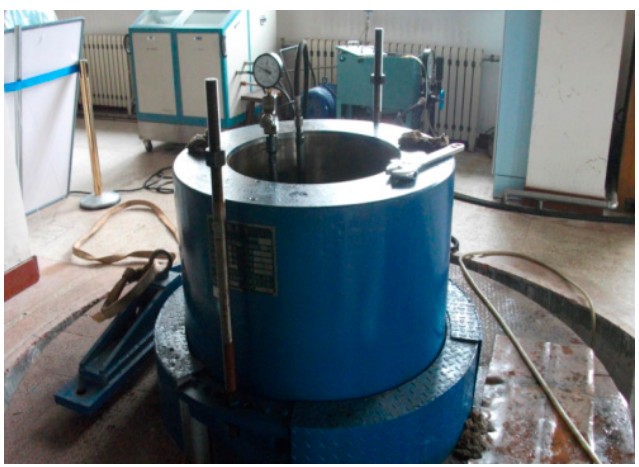

**Figure 12.** The pressure test chamber.

The test procedure was:

(1)    We placed the machine into the pressure chamber and installed the cover;
(2)    We first slowly increased the pressure to 10 MPa and held for 45 min, then slowly raised the pressure to 15 MPa for a period of 30 min, and then slowly increased the pressure to 20 MPa for a period of 30 min;
(3)    Pressure was relieved;
(4)    We removed the equipment to check for collapse or seal failure and checked if the sensor and winch operated normally;

The results showed that high-pressure resistance meets the design requirements.

## 5. Underwater Measurement Test

The purpose of this experiment was to test the underwater integrated measurement performance. The test venue was the Harbin Engineering University experimental pool of the National Key Laboratory of Science and Technology of Military Underwater Intelligent Robot Technology. The maximum depth was 10 m and the pool can simulate complex environments such as water movement and wave fluctuation.

The test procedure was as follows:

(1)    After the device was calibrated on land, the device was moved to a certain direction to be fixed, and a set of relative pose data values of the two pipelines were measured;
(2)    We maintained the electrical status on the system to ensure that the relative pose of the two pipelines was fixed and hang the entire experimental platform in the pool (Figures 13 and 14).

(3)　After the experiment platform was placed at the bottom of the tank, we adjusted the pool to maintain a certain flow, then we tensioned the test rope through the monitoring interface. The pose data are provided in Table 4. The results were compared with those of onshore experiments; and

(4)　After confirming the experimental data precision met the requirements, we lifted the experimental platform up and then the water tank experiment completed.

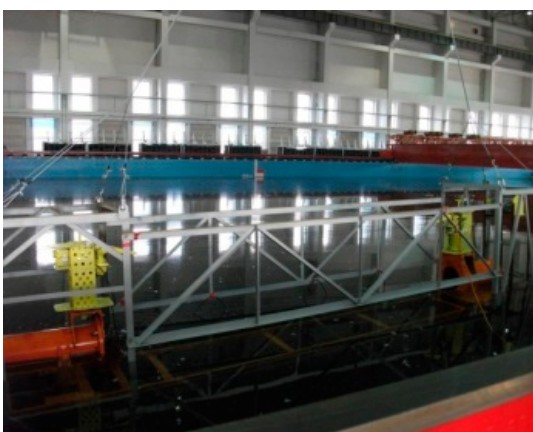

**Figure 13.** Hanging the experimental platform into the pool.

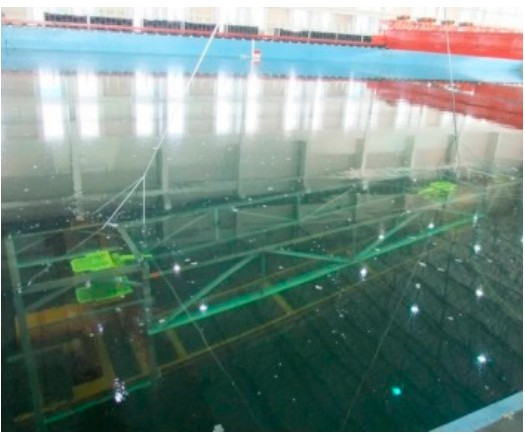

**Figure 14.** Measuring the experimental platform underwater.

**Table 4.** Underwater measurement results.

| Sequence | Measuring Equipment I | | | | Measuring Equipment II | | | | Rope Length (mm) |
|---|---|---|---|---|---|---|---|---|---|
| | $\alpha_r$ (°) | $\beta_r$ (°) | $\gamma_r$ (°) | $\theta_r$ (°) | $\alpha_b$ (°) | $\beta_b$ (°) | $\gamma_b$ (°) | $\theta_b$ (°) | |
| Measurement on land | −2.03 | 1.40 | 0.84 | −0.26 | 2.14 | −1.64 | −0.09 | 0.40 | 4531.7 |
| Measurement under water | −1.99 | 1.39 | 0.97 | −0.04 | 2.13 | −1.66 | −0.22 | −0.18 | 4531.3 |
| Difference | −0.04 | 0.01 | −0.13 | −0.22 | 0.02 | 0.02 | 0.13 | 0.57 | 0.4 |

According to Table 4, we determined the different underwater and onshore measured values of the device: the absolute value of the direction was less than 1° and the distance was less than 1 mm, indicating that the winch tension meets the measurement accuracy design requirements.

## 6. Conclusions

Based on the design of a measuring system that measures the distance and direction of two pipelines 1500 m deep in water, a pulling rope winch was designed. Combined with theoretical analysis, simulation and prototype tests produced the following conclusions:

(1) The device can fulfill the distance and measurement accuracy requirements for two submarine pipelines, and provide length and direction parameters for connecting pipelines.

(2) The performance of the winch is directly related to the testing accuracy of the device. The magnetic coupling coupler provides the winch with stable tension so the measurement device can achieve the desired measurement accuracy, which verifies the design.

(3) The magnetic coupling coupler improves the seal structure of the winch, which converts the dynamic seal into a static seal so that the winch performs well in deep water and has a simple structure.

**Author Contributions:** The first author, Z.W., conceived the framework of the article and wrote the article; the second author, D.L., designed the measurement device and the structure of the winch; the third author, T.W., made a pull rope winch tensile test for the motor control simulation; the fourth author, B.Z., was responsible for the underwater measurement experiment; All authors have read and agreed to the published version of the manuscript.

**Funding:** This paper was funded by NSFC (Contract name: Research on ultimate bearing capacity and parametric design for the grouted clamps strengthening the partially damaged structure of jacket pipes). (Grant number: 51879063) and (Contract name: Research on analysis and experiments of gripping and bearing mechanism for large-scale holding and lifting tools on ocean foundation piles), (Grant number: 51479043).

**Conflicts of Interest:** The authors declare no conflict of interest.

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
