# Peer review of "Design of a Measuring Device and Winch Structure for Detecting the Distance and Direction of Two Seabed Pipelines"

_jmse, doi:10.3390/jmse8020130_

Round 1

Reviewer 1 Report

The paper is very well written. The introduction is brief, but complete, covering background and literature review regarding the various technology involved in measuring distance and position of pipelines, with focus on current sealing methods. The design section clearly covers all parameters and equations involved in the proposed design; each variable is also quantified. The simulation section describes a standard brushless motor with position feedback, however the experimental section doesn’t verify the simulation specifically, but the overall design of the system.

Overall the design of the major components of a measuring system that measures distance and position of two pipelines at 1500 m deep-water is clearly described. Although the engineering design and methodology are advanced and certainly meet high standards, this paper seems to be more suited for a peer reviewed technical communication article than for a research journal. This is primarily because the novelty of the work is not obvious. On a similar note, the title “Research on Device Used to ….” should really be changed to: “Design of Device Used to…” or something along those lines.

Author Response

I am very grateful to your comments for the manuscript. According you’re your advice, we amended the relevant part in manuscript. Some of your questions were answered below.

To Reviewer #1:

Opinion 1-1: The paper is very well written. The introduction is brief, but complete, covering background and literature review regarding the various technology involved in measuring distance and position of pipelines, with focus on current sealing methods. The design section clearly covers all parameters and equations involved in the proposed design; each variable is also quantified. The simulation section describes a standard brushless motor with position feedback, however the experimental section doesn’t verify the simulation specifically, but the overall design of the system.

Responses to (1-1): According to this opinion, in the third part of this paper, based on the simulation analysis, we added the winch pull rope force test experiment, according to the design technical requirements, determined the winch output torque. Please see lines No.: 216-232 for details

Opinion 1-2: Overall the design of the major components of a measuring system that measures distance and position of two pipelines at 1500 m deep-water is clearly described. Although the engineering design and methodology are advanced and certainly meet high standards, this paper seems to be more suited for a peer reviewed technical communication article than for a research journal. This is primarily because the novelty of the work is not obvious. On a similar note, the title “Research on Device Used to ….” should really be changed to: “Design of Device Used to…” or something along those lines.

Responses to (1-2): According to this review, we have revised the title of the article “Design of a Measuring Device and Winch Structure for Detecting the Distance and Direction of Two Seabed Pipelines”. On this basis, other parts of the article are modified accordingly, as shown in red font.

************************************************

We would like to express our great appreciation to you and reviewers for comments on our paper. Looking forward to hearing from you.

Thank you and best regards.

Yours sincerely,

Bo Zhang

Zhuo Wang

Eail: [email protected]

Feb.07.2020

Reviewer 2 Report

In this paper an attempt is made to solve the problem of measuring the distance and position between the flanges of two pipelines on the seabed. In this end, a measuring device for pulling rope in seawater was designed. Aiming at the sealing problem of the key equipment that is the rotating shaft of the rope winch, this paper used the principle of the magnetic coupling to transfer the driving moment, and adopted the method of converting the dynamic seal into the static seal structure to realize the reliable sealing of the motor. Through the experiment of measuring two pipelines with underwater rope pulling device, it was verified that the measuring accuracy of the device meets the design requirements, and the feasibility of the application of magnetic coupling technology in winch is also verified.

This is an excellent paper with a novel approach and should be published. Some linguistic improvement will be needed before having it published.

Author Response

I am very grateful to your comments for the manuscript. According you’re your advice, we amended the relevant part in manuscript. Some of your questions were answered below.

To Reviewer #2:

Opinion 2-1: In this paper an attempt is made to solve the problem of measuring the distance and position between the flanges of two pipelines on the seabed. In this end, a measuring device for pulling rope in seawater was designed. Aiming at the sealing problem of the key equipment that is the rotating shaft of the rope winch, this paper used the principle of the magnetic coupling to transfer the driving moment, and adopted the method of converting the dynamic seal into the static seal structure to realize the reliable sealing of the motor. Through the experiment of measuring two pipelines with underwater rope pulling device, it was verified that the measuring accuracy of the device meets the design requirements, and the feasibility of the application of magnetic coupling technology in winch is also verified.

Responses to (1-1): According to this review, we have revised the title of the article “Design of the Structure of Measuring Device and its Winch for Detecting the Distance and Direction of Two Pipelines on the Seabed”. On this basis, other parts of the article are modified accordingly, as shown in red font. Opinion 2-2: This is an excellent paper with a novel approach and should be published. Some linguistic improvement will be needed before having it published. Responses to (2-2): According to the comments of this review, we have checked the grammar of the article in detail. We have finished entrusting the editor of this periodical department to polish this article.

  ************************************************

We would like to express our great appreciation to you and reviewers for comments on our paper. Looking forward to hearing from you.

Thank you and best regards.

Yours sincerely,

Bo Zhang

Zhuo Wang

Eail: [email protected]

Feb.07.2020
